# Deciphering the Biology of Circulating Tumor Cells through Single-Cell RNA Sequencing: Implications for Precision Medicine in Cancer

**DOI:** 10.3390/ijms241512337

**Published:** 2023-08-02

**Authors:** Santhasiri Orrapin, Patcharawadee Thongkumkoon, Sasimol Udomruk, Sutpirat Moonmuang, Songphon Sutthitthasakul, Petlada Yongpitakwattana, Dumnoensun Pruksakorn, Parunya Chaiyawat

**Affiliations:** 1Center of Multidisciplinary Technology for Advanced Medicine (CMUTEAM), Faculty of Medicine, Chiang Mai University, Muang, Chiang Mai 50200, Thailand; santhasiri.or@cmu.ac.th (S.O.); patcharawadee.t@cmu.ac.th (P.T.); sasimol.ud@cmu.ac.th (S.U.); sutpirat_m@cmu.ac.th (S.M.); songphon_sutthittha@cmu.ac.th (S.S.); petlada.yong@gmail.com (P.Y.); dumnoensun.p@cmu.ac.th (D.P.); 2Musculoskeletal Science and Translational Research (MSTR) Center, Faculty of Medicine, Chiang Mai University, Muang, Chiang Mai 50200, Thailand; 3Department of Orthopedics, Faculty of Medicine, Chiang Mai University, Muang, Chiang Mai 50200, Thailand

**Keywords:** circulating tumor cells, tumor heterogeneity, single-cell RNA and transcriptome sequencing, personalized and precision medicine, metastasis, drug resistance

## Abstract

Circulating tumor cells (CTCs) hold unique biological characteristics that directly involve them in hematogenous dissemination. Studying CTCs systematically is technically challenging due to their extreme rarity and heterogeneity and the lack of specific markers to specify metastasis-initiating CTCs. With cutting-edge technology, single-cell RNA sequencing (scRNA-seq) provides insights into the biology of metastatic processes driven by CTCs. Transcriptomics analysis of single CTCs can decipher tumor heterogeneity and phenotypic plasticity for exploring promising novel therapeutic targets. The integrated approach provides a perspective on the mechanisms underlying tumor development and interrogates CTCs interactions with other blood cell types, particularly those of the immune system. This review aims to comprehensively describe the current study on CTC transcriptomic analysis through scRNA-seq technology. We emphasize the workflow for scRNA-seq analysis of CTCs, including enrichment, single cell isolation, and bioinformatic tools applied for this purpose. Furthermore, we elucidated the translational knowledge from the transcriptomic profile of individual CTCs and the biology of cancer metastasis for developing effective therapeutics through targeting key pathways in CTCs.

## 1. Introduction

Circulating tumor cells (CTCs) mediate metastasis through the dissemination of cancer cells into the bloodstream, referred to as the “seed” of metastases [1]. Once entering the systemic circulation, CTCs may have their cellular and molecular characteristics altered to survive in the harsh environment of the bloodstream, which includes a plethora of circulating immune components [2]. Blood microenvironments such as fluid sheer stress and mechanical squeezing in constricted vessels can induce biochemically mediated molecular and morphological changes of the CTCs, for example, through the epithelial-mesenchymal transition (EMT) process [3]. The EMT process is the transdifferentiation of epithelial cells by which the cells lose their epithelial characteristics and acquire mesenchymal features [4]. Cancer cells hijack these dynamic changes in morphology and motility to conduct cancer migration, invasion, and intra- and extravasation into the circulatory system. The EMT process also activates several survival pathways to avoid anoikis and induce resistance to chemotherapy and physical stress [5,6]. The EMT phenotype of CTCs is to survive in the bloodstream until undergoing mesenchymal-to-epithelial transitions (MET) and restoring their epithelial character to extravasate and colonize distant sites [7].

CTCs encompass a tremendously heterogeneous population of cells that reflects the diversity within the tumor, in which only certain cells are conduits to metastasis. Metastasizing populations of CTCs possess distinct biological characteristics to allow themselves dissemination, survival, and seeding as a consequence of genetic and transcriptional alteration in genes associated with metastasis [1]. Furthermore, the hostile microenvironment of CTCs is a complex network involving various cell types and soluble components [8,9]. The phenotypes of CTCs might be altered by biochemical regulatory adaptations and interactions with blood components during transition through the microcirculation that induce adaptations in cellular deformation and stiffness [10,11]. The interactions of CTCs with hematopoietic cells and stromal cells also occur to withstand the physical stress and stabilize arrest [12].

The long-standing view of the metastatic cascade has paved the way for the development of cancer therapy that inhibits metastasis at specific steps of the metastatic cascade, including targeting the initial steps of mobilization, invasion, and intravasation of the primary tumor into the blood or lymphatic circulation, the intermediate steps of transition within the vasculature and extravasation, and the late step of colonization [13]. Regarding anti-metastasis therapy, two key strategies are the prevention of cancer spreading and the suppression of pre-existing metastases. To prevent cancer cells from metastasizing to distant sites, precise decision-making is required before prescribing anti-metastasis drugs to patients. This necessitates the development of a potential marker to stratify high-risk patients and minimize overtreatment in low-risk patients who are prone to being cured by standard treatment. Detecting CTCs in blood circulation is the most promising companion marker and new therapeutic target in this respect. Anti-metastatic treatment of CTCs, which are the seeds of metastasis, would establish an effective strategy towards preventing or eliminating the expansion of secondary lesions and lead to a curative effect in cancer patients. Therefore, the molecular mediators of CTC dissemination and self-seeding should be characterized.

To explore the CTC journey towards metastasis, it requires high-resolution technology to dissect the complexity of tumor cells shedding in the circulatory system and identify the driving mechanism of the CTC invasion and the interaction of CTCs with other circulating cells [14]. With the advancement in next-generation sequencing (NGS) and bioinformatics, we can gain a better understanding of the molecular mechanisms of complex metastastic processes through the genotypic and phenotypic characterization of CTCs. Single-cell RNA sequencing (scRNA-seq) technology is the most powerful tool to investigate gene expression profiles at single-cell resolution that bear great potential to reveal key regulatory pathways behind metastases.

Many attempts have been made to investigate the molecular characteristics of CTCs through a developed workflow that effectively enriches and isolates the CTCs from a vast majority of blood cells, with the ultimate aim of capturing most of the CTC populations. Together with the advanced scRNA-seq platforms, the metastatic mechanisms of cancer have been unveiled, paving the way for novel treatment and intervention. In this review, we summarize the updated and promising workflows for CTC enrichment, isolation, and characterization through scRNA-seq technology. Furthermore, regarding the examination of single CTC transcriptome analyses, the remarkable value of the single CTC analyses and the corresponding validation are also discussed.

## 2. Workflow of scRNA-Seq of CTCs

### 2.1. Single-Cell Isolation Techniques

CTCs are extremely rare, found with only one cell per billion blood cells in circulation, either as a single unit or a multicellular grouping [15]. Although, on average, only a small number of CTCs can be obtained from blood samples, the number of CTCs detected in cancers varies between different tumor types, tumor sizes, stages, or even in independent cohorts [16,17], as shown in Table 1. Different workflow and sample handling methods applied for CTC enrichment also significantly affect CTC recovery, including pre-analytical blood processing methods, time of processing, enrichment platforms, and the variety of CTC detection markers [18]. The CellSearch^®^ system has been approved by the Food and Drug Administration as the first CTC enumeration test for predicting overall and progression-free survival as well as to aid clinical decisions in patients with metastatic breast, prostate, and colorectal cancer [19]. However, the limitations of the CellSearch^®^ system, which uses epithelial cell adhesion molecules (EpCAM) as a marker for CTC epithelial lineage, are an underestimation of the EpCAM^low/−^ CTC population, which may not represent the CTC heterogeneity [20,21]. In this multiple cohort study, the number of CTCs can be detected at 61 ± 696 (means ± SD) cells in 7.5 mL blood samples of metastatic carcinomas [22]. The highest number of CTCs was detected at 84 ± 885 cells in breast cancer, whereas only 1 ± 1 (means ± SD) CTC was detected in renal cancer after enrichment.

A range of isolation methods are currently applied to capture the broad CTC population for further investigation of the CTC heterogeneity that will give more insight into the metastatic mechanism of cancer [23]. In this respect, to avoid the bias of surface markers, a variety of CTC enrichment platforms have been developed using the size, density, electrical charges, and other physical characteristics of CTCs (Figure 1). These label-free methods have been widely applied and are compatible with a downstream scRNA sequencing analysis of CTCs.

After the CTC enrichment step, a proportion of white blood cells are still contaminating the CTC population, so more purity of CTCs is required for further investigating the phenotype and genotype of CTCs at the single cell level. The methods for single-cell isolation, including limiting dilution, fluorescent-activated cell sorting (FACS) isolation, micromanipulation, laser capture microdissection (LCM), and microfluidic technology, have been developed to effectively isolate an intact single-CTC from a complex blood component (Figure 1).

#### 2.1.1. Fluorescent-Activated Cell Sorting (FACS) Isolation

By means of FACS systems, target populations of CTCs tagged with a fluorescent marker, such as a fluorophore-conjugated antibody, are separated by flow cytometry. As cell suspensions are driven through the cytometer, each CTC rapidly passes to a laser, which subsequently allows optical detectors to identify cells based on the designated characteristics [14]. High-throughput CTC enrichment connected directly to a FACS sorter is developed to isolate and sort single CTCs into microplates containing molecular indexing and sample barcoding for transcriptomic profiling in human prostate cell lines [24]. This rapid one-step process allows >90% post-enrichment recovery of viable CTCs that is compatible with downstream RNA analysis at the single-cell level. A single-cell analysis of sorted CTCs was reported in a preclinical model of pancreatic cancer [25]. Identifying subpopulations of CTCs expressing epithelial and mesenchymal genes showed an obvious similarity to their surface protein expression. However, the limitations of the method are due to the requirement of a high number of input cells (at least 10,000 cells in suspension), which might not be suitable for isolating low-quantity cell populations of CTCs [26].

**Table 1 ijms-24-12337-t001:** Overview of CTC enrichment method.

Enrichment Based Methods	Publication	Blood	Samples (Number and Characteristics)	Enrichment Technology	Enrichment Features	Capture Efficiency	Reference
Breast cancer							
EpCAM	Talasaz et al., 2019	A total of 9 mL	A total of 17 metastatic cancers	MagSweeper	anti-EpCAM immunomagnetic separation	In total: 12 ± 23 (means ± SD) CTCs per 9 mL.	[27]
	Riebensahm et al., 2019	A total of 7.5 mL	A total of 44 brain metastases in breast cancers	CellSearch	anti-EpCAM immunomagnetic separation	In total: 1 to 1800 with median number of 4 CTCs per 7.5 mL blood.	[28]
	Nagrath et al., 2007	A total of 0.9–5.1 mL	A total of 10	CTC-Chip	anti-EpCAM coated microposts based microfluidic	In total: 5 to 176 with 79 ± 52 (mean ± SD) CTCs per mL.	[29]
Non-EpCAM	Drucker et al., 2020	A total of 15 mL	A total of 28 metastatic breast cancer	RosetteSep™	Immunodensity with negative depletion of WBCs	In total: 0.55 CTCs/mL (mean).	[30]
			A total of 8 metastatic breast cancer	ScreenCell^®^ Cyto filters	Size-based methods	mean count 4.2 CTCs per mL.	
Colorectal cancer (CRC)							
EpCAM	Nagrath et al., 2007	A total of 0.9–5.1 mL	A total of 10	CTC-Chip	anti-EpCAM coated microposts based microfluidic	In total: 42 to 375 with 121 ± 127 (mean ± SD) CTCs per mL.	[29]
	Tsai et al., 2016	A total of 2 mL	non-metastatic (*n* = 95), and m-CRC (*n* = 15) patients.	CellMax (CMx^®^)	anti-EpCAM coated chip	In total: 19 ± 53 CTCs per 2 mL for non-metastasis; In total: 51 ± 147 CTCs per 2 mL for metastasis.	[31]
	Dizdar et al., 2019	A total of 7.5 mL	A total of 80 CRC with M0 and M1	CellSearch	anti-EpCAM immunomagnetic separation	A total of 135 CTCs from *n* = 80.	[32]
	Wu et al., 2017	A total of 7.5 mL	A total of 44 CRC	CellSearch	anti-EpCAM immunomagnetic separation	In total: 0 to 8 with mean count of 2 CTCs per 7.5 mL for stage I–II; In total: (0–35) with mean count 2 CTCs per 7.5 mL for stage III–IV.	[33]
Non-EpCAM	Hendricks et al., 2020	A total of 8 mL	A total of 31 diagnosed with colon carcinomas and rectal carcinomas	ScreenCell^®^ Cyto filters	Size-based methods	In total: 0.2 to 14.3 with mean count 3.25 CTC per mL.	[34]
	Vasantharajan et al., 2022	A total of 8 mL	A total of 15 CRC with various stages of cancer (AJCC stages I-IVB)	MetaCell	size-based separation	In total: 0 to 12 with mean count of 2 CTCs per 7.5 mL.	[35]
Hepatocellular carcinoma (HCC)							
EpCAM	Morris et al., 2014	A total of 7.5 mL	A total of 50 HCC	CellSearch	anti-EpCAM immunomagnetic separation	In total: 0 to 8 CTCs per 7.5 mL.	[36]
	Zhang et al., 2016	A total of 2 mL	A total of 36 HCC	CTC-chip	anti-EpCAM coated microposts based microfluidic	In total: 14 ± 10 (mean ± SD) CTCs per 2 mL.	[37]
Non-EpCAM	Morris et al., 2014	A total of 7.5 mL	A total of 50 HCC	ISET	size-based separation	In total: 0 to 47 CTCs per 7.5 mL.	[36]
	Zhao et al., 2023	A total of 7.5 mL	A total of 127 HCC patients With low recurrence (LR) and high recurrence (HR)	Leucosep^®^ with CD45 depletion	Density and immunomagnetic separation with negative depletion of WBCs	In total: 0 to 20 with means of 8 CTCs per 7.5 mL for LR-HCC; In total: 0–34 with means of 12 CTCs per 7.5 mL for HR-HCC.	[38]
Lung cancer							
EpCAM	Nagrath et al., 2007	A total of 0.9–5.1 mL	A total of 55 NSCLC	CTC-Chip	anti-EpCAM coated microposts based microfluidic	In total: 5 to 1281 with 155 ± 236 (mean ± SD) CTCs per mL.	[29]
	Ke et al., 2015		A total of 7 NSCLC	NanoVelcro	anti-EpCAM)-coated nanostructured substrates in microfluidic chip	In total: 2 to 7 with 7 ± 4.74 (mean ± SD) CTCs per mL.	[39]
Non-EpCAM	Hosokawa et al., 2013	A total of 7.5 mL	NSCLC	miniaturized microcavity array (MCA) system	size-based separation	In total: 0 to 291 with median of 13 CTCs per 7.5 mL.	[40]
	Sonn et al., 2017	A total of 5 mL	A total of 66 stage I–III patients and 16 stage IV	ISET	size-based separation	1.48 ± 1.71 CTCs per 5 mL for stage I–III patients; 8.00 ± 9.95 per 5 mL for stage IV patients.	[41]

#### 2.1.2. Micromanipulation

The micromanipulator is operated through microscope-guided manual capillary pipettes. CTCs can be selectively separated and transferred with high efficiency. Micromanipulator technology for the isolation of single CTCs has been developed as automated robotic platforms, including the CellCollector, CytePicker, and semi-automated single-cell aspirator. Briefly, the system is a combination of high-speed scanning microscopy with imaging processing software for rare-cell detection and an automated micromanipulation robot. Each retrieved single cell can be isolated by visualization under the microscope, thus enabling high-precision and unbiased isolation.

The CellCollector system employs stainless steel wires that are functionalized with antibodies against EpCAM to isolate CTCs directly from blood samples and decollate them for single-cell downstream analysis [42].

The CytePicker module utilizes a durable needle tip to mechanically dislodge individual CTCs from microscope slides [43]. This single-cell retrieval device is normally combined with the RareCyte/CyteFinder instrument that allows for CTC separation and identification by imaging analysis, thereby allowing for single-CTC recovery with minimal target-cell loss [44]. The detection and retrieval of spiked CTCs by the integrated assay showed no obvious effect on their transcriptome after being analyzed by scRNA-seq [45].

SASCA, a seamless integration of commercially available devices and custom engineered parts developed by Tokar et al., showed that the system had the capability to individually aspirate target CTCs from a contaminated background of blood cells using a multi-axis micromanipulator [46].

The operation of micromanipulators is mainly controlled by sophisticated software to minimize user technical skill and take a few minutes to process single-cell isolation. These platforms are, therefore, sensitive and compatible for comprehensive analysis of cancer cell-spiked samples and clinical specimens with a purity level exceeding 90%. Those isolated CTCs exhibited high quality for single-cell-based molecular analysis such as comparative genomic hybridization [47] and NGS such as whole genome/whole exome sequencing [46,48,49,50] and RNA sequencing [45,46]. In this regard, a more inclusive study of the performance of a scRNA-seq-based micromanipulation system in clinical specimens should be observed. Although micromanipulators show a very promising approach for retrieval of single cells, an example study performed on blood samples of hepatocellular carcinoma (HCC) patients showed that 40% of single CTC recovered by robotic micromanipulators were qualified for further scRNA-seq analysis [51].

#### 2.1.3. Laser Capture Microdissection (LCM)

LCM is a premier technology for harvesting pure single cells from mostly complex samples on a microscope slide. Similar to the micromanipulation method, the basic principle begins with observing the cell of interest based on morphological or phenotypic characteristics via a visual microscope. Subsequently, instead of using a micropipette, single cells are dissociated by a focused laser pulse to capture the cells for further analysis [52].

LCM has been used in combination with the CTC enrichment method to provide ultra-high purity of single-cell isolation that is amenable to biological characterization by NGS. As with the NanoVelcro chip, an existing microfluidic for CTC enrichment integrated with LCM showed that the high quality of single CTCs was successfully harvested. NanoVelcro-LCM technology is not only capable of capturing CTCs but is also compatible with performing Sanger sequencing and NGS [53,54,55,56].

Single CTCs in melanoma patients have been characterized by genotyping and contain a signature oncogenic mutation, such as BRAF^V600E^. Most importantly, the BRAF^V600E^ mutation was found in single CTCs matched to the traditional melanoma biopsies that were detected by the conventional PCR-based technique [53]. Similarly, using high-quality whole genome sequencing on prostate CTC, the individually isolated cells represented a quality comparable to their tissue sequencing with more than 90% coverage. The comparison of single nucleotide polymorphisms was not different between CTCs and their tissue counterparts [54]. Moreover, approximately 25% to 80% of targeted exome regions were detected in a single CTC with no chromosomal loss after establishing all processes [55]. As a proof-of-concept, NanoVelcro-LCM demonstrates the potential for a streamlined workflow to accurately analyze the genetic material of CTCs at the single nucleotide level.

#### 2.1.4. Microfluidic Technology

Microfluidic systems have been known as a powerful platform for single-cell processing on a high-throughput scale, capable of analyzing complex samples such as organs, tissues, and whole blood samples. A scRNA-sequencing-based microfluidic device enables single-cell isolation, RNA extraction, barcoding, and library preparation in one sample microfluidic device [57]. This technique has been applied for single-cell isolation of CTCs, in which the enriched sample is isolated through the use of a droplet-based microfluidic system [58,59]. A high-throughput single-cell platform, 10X Genomics, has been applied for the identification and characterization of CTCs in HCC patients [59]. A total of three CTCs were identified from a total of 3000–7000 cells in enriched samples from two candidate patients. Approximately 30–60% of input cells were partitioned and recovered, meaning that at least 50% of the input cells were lost in each sample [60]. Considering the rarity of CTCs in blood, with an estimated number of 1–10 per mL of blood, this approach might have a limitation for retrieving single cells from a small cell population [61]. Therefore, high sensitivity and high recovery methods are required in the single-cell transcriptome analysis workflow.

Hydro-Seq has been applied for single-cell transcriptome analysis of 666 CTCs isolated from 21 breast cancer patients [62]. By integrating microfluidic circuit design with valve controls, CTCs are arrested at the cell capture site while the smaller blood cells pass through. Within the same chamber, a single CTC can be barcoded and continued for the rest of the process, including PCR amplification, library construction, and sequencing using the Drop-seq protocol. Hydro-seq represents high CTC capture efficiency and reproducibility. The number of CTCs was recovered at a high cell capture efficiency of 72.85 ± 2.64% (mean ± SD), and gene expression profiles are comparable with the results from biological replicates of blood samples from the same patients. Additionally, a low percentage of cells expressing CD45 leukocyte markers and hemoglobin was found in the scRNA-seq data. The results from gene expression analysis confirm the efficacy of removing massive contamination of blood cells in a single-cell capture chamber.

Similarly, microfluidic chips for single-cell RNA sequencing (SCR-chip) are recognized as “whole blood in, single-CTC RNA out” [63]. The workflow also includes the screening section, where labeling CTCs with immunofluorescence is analyzed by automated computer algorithms before single-CTC lysis. Subsequently, the RNA product is released and directly used for scRNA-seq according to the SMART-seq II sequencing protocol. The spiking experiments showed that an affordable quality of individual cells isolated from peripheral blood without leukocyte contamination could be obtained by SCR-chip [41]. The CTC enrichment efficiency of an integrated microfluidic chip with low to high cell numbers (10 to 2000) of the MCF-7 cell line varies between 55 and 93%, which is relatively dependent on the adjusted flow rate. The RNA yields and quality obtained from trapped single-cell CTCs are adequate for RNA sequencing and transcriptomic analysis.

In vivo optofluidic platforms have been established with the additional feature of microfluidic cell sorting chips for real-time isolation of CTCs from genetically engineered mouse models [64]. CTCs are collected during the continuous flow of blood from living mice into the system. The microfluidic valves trap CTCs as the cells flow through the device, and CTC-depleted blood is returned back to the mouse via the shunt. After sampling, CTCs are further enriched by a secondary single-cell CTC sorting chip to flush individual cells into microwells for downstream scRNA-seq. These studies suggest that integrated microfluidic systems for one-step enrichment and single-cell analysis have been devised to increase CTC recovery.

With the development of functionally integrative technology, the efficiency of CTC recovery and quality of genomic material have been improved, which greatly promotes the study of CTCs at single-cell resolution. The enrichment platform that is suitable for further studying scRNA-seq is summarized in Figure 1.

### 2.2. Data Analysis in scRNA-Seq

One important application of scRNA-seq studies is the use of computational algorithms to accurately detect the number of cell types in a sample. Several analytical challenges remain due to the unique characteristics of scRNA-seq data, including the massive amount of single-cell data, biological variability, data sparsity, technical noise, and dropout events [65]. Like bulk RNA-seq, a universal standardization of computational methods has been developed for scRNA-seq analysis that includes cell clustering, the transcriptome landscape of differentially expressed genes, and cell trajectories, as shown in Figure 1.

#### 2.2.1. Cellular Subpopulation Identification

Identification of discrete cellular subpopulations is one of the exciting areas for scRNA-seq experiments. Cell clustering begins with evaluating cell-cell similarity and determining a certain number of segregated clusters through unsupervised learning [66]. The main challenges of the clustering technique are that the cells presented in the data are highly heterogeneous and need to be determined by cell type at various stages. Therefore, computational methods in this category focus on the dimensional reduction of high-dimensional data to aid data interpretation. Principal component analysis (PCA), t-distributed stochastic neighbor embedding (t-SNE), and uniform manifold approximation and projection (UMAP) are currently the most used methods [67]. Due to the non-linear structure of scRNA-seq data, such analysis poses high levels of technical noise from outlier cell populations and dropout events, making the scRNA-seq data highly noisy and sparse [68]. Although PCA is a favorable linear dimension-reduction method, it is incapable of analyzing large-scale scRNA-seq and visualizing the data [69]. Instead, t-SNE and UMAP are non-linear dimensionality reduction methods that extract a low-dimensional representation while retaining high-dimensional data structure similarity [70,71].

Other novel methods based on deep learning have recently emerged, such as scRNA-seq deep embedding clustering via convolutional autoencoder embedding and soft K-means (scCAEs) [72] and single-cell model-based deep embedded clustering (scDeepCluster) [66]. Additionally, the methods feed forward to combine with neural networks, such as single-cell variational inference (scVI) [73], variational autoencoder (VAE) [74], deep count autoencoder (DCA) [75], and deep embedding algorithm for single-cell clustering (DESC) [76]. These methods hold promise as they are capable of capturing complex patterns in high-dimensional scRNA-seq data while also addressing challenges such as denoising of single-cell transcriptomes, batch effect removal, and probabilistic modeling of cell type clustering [77]. An example of such work provided by CTCs clustering is called deep dictionary learning using k-means clustering cost (DDLK). DDLK incorporates k-means clustering into deep dictionary learning to precisely yield groupings of CTC and WBC populations by using pathway enrichment scores at the single-cell level [78]. Notably, cell clustering methods are progressively being established to surpass the existing methods for handling assorted scRNA-seq data. Each clustering algorithm in scRNA-seq analysis may vary depending on the specific research question and the characteristics of the dataset. With this respect, it is challenging to identify clusters of rare cell populations of CTCs with an occurrence rate of less than one in a hundred or thousand cells. Accumulating evidence performed unsupervised hierarchical clustering and successfully accessed rare cell types as small clusters [79]. Many clustering methods with a progressive design for rare cell population-specific genes have been extensively developed with the need for computational scalability and flexibility.

#### 2.2.2. Differential Gene Expression Analysis

The next step in the workflow of scRNA-seq is the differential gene expression (DGE) analysis, where the genes are identified individually for expression differences. DGE analyses provide insight into the genetic mechanisms behind the phenotypic variations associated with tumor genesis [65]. Several methods were originally developed for detecting DEG based on bulk RNA-seq data, and those can be applied to scRNA-seq data, such as DESeq, edgeR, Limma, and SAMseq [80]. However, scRNA-seq data typically differs from conventional bulk RNS-seq in several aspects. ScRNA-seq has a higher level of technical noise than bulk-cell data due to the tiny amount of input mRNA from single cells, amplification biases, dropout phenomena, and stochastic bursting events [81]. To tackle these issues, a number of software packages that are specifically designed for single-cell data have been proposed, for instance, the beta-Poisson model for scRNA seq (BPSC) [82], model-based analysis of single-cell transcriptomics (MAST) [83], Monocle [84], and DEsingle [85]. These methods are tailored to handle the unique characteristics of scRNA-seq data and provide more accurate and robust differential gene expression analysis at the single-cell level. Each DGE tool employs different statistical tests with quantitative changes in transcriptional differences between cells, allowing for the elucidation of differentially expressed genes in each cell subpopulation, specific condition, or stage. The comparative analysis of DGE analysis has shown fundamental differences in terms of the number of detected differentially expressed genes, false-positive genes, and accuracy, which vary among the compared methods [86,87]. However, it is currently unclear whether the established methods are the most suitable for scRNA-seq analysis. The appropriate method can be selected based on the evaluation criteria for a specific application.

The specifically designed methods for scRNA-seq analysis do not outperform those conventional bulk RNA-seq analysis methods [88,89]. For highly expressed genes, many bulk-RNA-seq tools can deliver similar results to the approaches created for scRNA-seq datasets. On the other hand, the detection results are shown to vary for lowly expressed genes [90]. Considerably, the degree of gene expression is crucial for determining the performance of DGE methods. In these cases, the comparing statistical analysis method for scRNA-seq can serve as a reference for choosing the most suitable method for single-CTC RNA sequencing data.

In addition, an unbiased and accurate annotation of rare cell subpopulations is challenging. There are several factors that influence the power of the analysis of DEG detection across different conditions as well as across cell types [91]. Given that many aspects, such as sequencing depth, dropout rates, cell population proportion, the number of cells, and biological replicates, as well as multiple testing methods, can affect the DEG analysis, for example, Sun et al. successfully examined the gene profile of CTCs from four vascular-specific sites using the edgeR package for DEG detection [51]. Only genes expressed in at least 90% of the samples in the group were selected as potential DEGs. Another study was exemplified by Brechbuhl et al., who performed a scRNA-seq dataset with Seurat (version 3), and the analysis required those genes to be detectable in 25% of cells for at least one of the compared groups [92].

#### 2.2.3. Pseudotime Analysis

Single-cell-based trajectory inference, or pseudotime analysis, aims to map a set of high-dimensional transcriptomic data from a cross-sectional cohort of individuals. Pseudotime analysis infers the ordering of cell states based on their similarity in gene expression, allowing for the determination of dynamic cellular processes experienced by individual cells. This approach can be used to depict cellular differentiation or relative progression [93]. In this respect, it can precipitate a change in CTC status and segregate molecular programs of cell fates underlying EMT plasticity or their trafficking into the bloodstream along the time (e.g., untreatment vs. treatment) or space (e.g., anatomical blood flow) of blood sampling. These may be applied to uncover the nature of the cell transition program that leads to defining the role of CTCs in metastasis.

The computational methods are built on the premise that cells with similar gene expression patterns have closer developmental paths. Although many benchmarks for trajectory analysis have been developed, a typical workflow for these methods is designed into three main steps: (i) preprocessing data by filtering out the poor-quality cells and uninformative expressed genes, normalization to remove undesired technical or biological errors, and dimensionality reduction; (ii) clustering the cells by determination of global lineage structure; and (iii) pseudotemperal reconstruction by projection of respective cells onto their lineage individuals along time within one biological sample [94]. In a comprehensive statistical framework, the analysis uncovers multiple aspects, including (i) the genes in which expression is associated with programmed lineages along differentiation, (ii) differential gene expression between the transcriptional lineages over pseudotime, and (iii) a density or proportion of cells along lineage across conditions [95]. Take Monocle as an example: it represents the expression profile of each cell by dimensionality reduction in an independent component analysis (ICA). A minimum spanning tree (MST) is constructed to “join the dots” between transcriptionally similar cells, and the longest path through this tree will be considered the default setting for differentiation [96]. Monocle 2 has been re-engineered to infer differences in gene expression patterns of transcriptional lineages with complex branching events using discriminative dimensionality reduction via learning a tree (DDRTree) and ordering cells by reverse graph embedding [97]. More recently, Monocle 3 has been developed to work well with large single-cell datasets and support trajectories with multiple roots. The algorithm is still being refined, as described elsewhere [98,99]. Other recent approaches have been adopted, e.g., Palantir [100], Slingshot [101], PAGA [102], STREAM [103], Tempora [104], and tradeSeq [97], but there has been little cross-over of methodological development.

Collectively, the ongoing improvement of scRNA-seq technology and concurrent advances in bioinformatics approaches can provide deeper insight into the gene expression heterogeneity and dynamic cellular processes in cancer tissues [105]. However, most scRNA-seq pipelines are not initially designed for the investigation of small populations of cells, such as CTCs in blood samples. To overcome the limitations of current scRNA-seq methodologies, sophisticated computational methods have been improved for single-CTC analysis. Schissler et al. (2016) developed a novel aggregation method and cell-centric statistics framework for analyzing scRNA-seq data of CTCs derived from prostate cancer patients [106]. The analytical approach simulated transcriptome dynamics to predict differentially expressed pathways (DEPs) by examining aggregated cell-cell statistical distances within biomolecular pathways. The novel method performed comparably to conventional methods of Gene set enrichment analysis (GSEA) and was superior to single-cell differentially expressed genes (SCDE) followed by gene set enrichment [106].

Teschendroff et al. (2017) further introduced a novel algorithm of single-cell entropy (SCENT) for discriminating CTCs based on their biological differentiation potency and phenotypic plasticity [107]. A scRNA-seq dataset of CTCs was derived from resistant prostate cancer patients exhibiting enzalutamide resistance during treatment. The computed signaling entropy rate (SR) obtained from CTCs in untreated patients was lower than that of those who developed resistance under treatment. The sorting of CTCs based on SR into low and high groups represented a positive correlation between SR value and the expression of ADLH7A1, a cancer stem cell (CSC) marker for enzalutamide resistance. This algorithm demonstrated the clinical utility of SR to be applied for identifying putative CSC and drug-resistant status [107].

## 3. Translational Relevance of scRNA-Seq of CTCs

### 3.1. Decoding Tumor Heterogeneity in CTCs and Their Gene Expression Signatures Implicating Clinical Outcome

Intra-tumoral heterogeneity (ITH) is one of the key features in the progression of invasive tumors and malignant transformation [108]. All the genetic variation, tumor microenvironment, and reversible changes in cellular characteristics influence the phenotypic and functional heterogeneity of cancer cells in the tumor mass. This ITH mainly relates to the clonal evolutionary fate that is the major cause of drug resistance, metastasis, and poor clinical outcomes [109,110]. Identification of a drug-resistant and metastatic subpopulation is, therefore, of the utmost importance for delineating the mechanism of disease progression and treatment implications.

The follow-up of the dynamic evolution of cancer requires repeated sampling throughout the treatment duration, which is not possible by performing a tissue biopsy. Analyzing individual CTCs in the blood samples is a promising approach. Compared with a single sampling of the primary tumor and a whole-tissue biopsy of metastases, scRNA-seq analysis of the CTC provides additional insight into the molecular landscape of tumor invasion and metastasis [111]. Here, we focus on the translational potential of scRNA-seq analysis of CTCs and the clinical implications of CTC heterogeneity analysis in a wide range of cancers.

#### 3.1.1. Prognostic Molecular Markers

Being able to predict clinical outcomes, treatment responses, and stratify high-risk groups is a significant approach to overcoming the delayed management of poorly prognostic patients. Analysis of sc-RNA sequencing of CTCs provides insightful phenotypic information about CTCs at various stages of the disease that mainly indicates the plasticity of CTCs with distinct phenotypes of epithelial, mesenchymal, or hybrid CTCs [112]. Distinct CTC populations, including EMT and mesenchymal-epithelial transition (MET)-like CTCs as well as CTCs with different receptor expressions (ER, PR, AR, and HER2), were detected in breast cancer patients by using the Hydro-Seq technique [62]. HER2-positive and HER2-negative CTCs expressed EMT- and MET-related genes, respectively. These findings indicate a relationship between the epithelial and mesenchymal status of CTCs and HER2 expression. Furthermore, a rare population of CTCs was detected, including those expressing the stemness markers of epithelial CSCs (ALDH) and mesenchymal CSCs (CD44+/CD24−) [62]. These findings suggest that different CSC regulation pathways are activated in EMT and MET-CTCs, which represents the extensively heterogeneous nature of breast CTCs.

CTC subtyping is a powerful prognostic indicator for various types of cancer. The presence of mesenchymal CTCs (VIM, SPARC, and ITGB1) in blood samples of colorectal patients exhibits a strong prognostic value for both progression-free survival and overall survival (Figure 2) [113]. The results are also in accordance with the study in prostate cancer patients with metastasis [114]. Interestingly, most of the CTCs with an epithelial phenotype were concomitantly found with their stem cell-like markers [113]. Several studies also suggest that the detection of the CSC phenotype during epithelial-mesenchymal plasticity serves as a prognostic and predictive tool for unfavorable clinical and treatment outcomes and identifies potential therapeutic targets [115,116,117]. Subgroup analysis of CTCs expressing EMT and stemness genes in advanced-stage HCC patients showed an association between the upregulation of *VIM* and *ROM1*, *POU5F1*, *NOTCH1*, and *STAT3* detected in CTCs and a poor prognosis (Figure 2) [118]. These CTC subgroups exhibited a canonical hepatocyte-related gene, indicating their hepatocyte origin. However, how the dynamic cellular plasticity expressing EMT traits is associated with the acquisition of CSC in CTCs and the mechanistic aspects of the co-expressed phenotype related to drug resistance remain inconclusive. 

#### 3.1.2. Treatment Response and Disease Progression

The scRNA-seq analysis of CTCs in the bloodstream potentially identifies the aberrant expression of genes involving the aggressiveness of CTCs that can be further applied for monitoring the therapeutic response and metastasis risk of cancer patients. By using scRNA-seq analysis of breast CTCs, the results showed the expression of the pre-adapted (PA) transcriptional state in breast CTCs, representing a quiescent subpopulation in primary tumors [119]. These cells acquired a resistant phenotype through transcriptional reprogramming and genetic alteration upon chemotherapy treatment. The expression pattern of the PA signature was dominant in CTC clusters with mixed epithelial and mesenchymal characteristics, given that PA-phenotypic CTCs may not only confer a survival advantage in the early stages of therapy but also initiate micro-metastatic spread [119].

CTC-derived estrogen receptor (ER) signaling can be used for early monitoring of treatment response. Analysis of scRNA-seq found that the 17 gene transcripts were specifically expressed in metastatic breast CTCs and that a 6-gene set (*PIP*, *SERPINA3*, *AGR2*, *SCGB2A1*, *EFHD1,* and *WFDC2*) was remarkably identified as a “resistance signature” and associated with rapid disease progression [120]. Likewise, the 18 prognosis-related genes retrieved by scRNA-seq exhibited a strong association with aggressive malignancy characteristics and poor survival outcomes in breast cancer [121]. Applying risk score construction, it was shown that a high risk score was related to poor survival and high metastasis risk. Because the AKT-mTOR and CDK pathways were activated in CTCs from high-risk patients, this patient group was more sensitive to AKT-mTOR and CDK pathway inhibitors than low-risk patients individuals. Therefore, the risk score construction from constituted genes uncovered by scRNA-seq had potential for precision targeted therapy. The designated drugs for treating those patients could alternatively target the abnormally activated genes and/or pathways.

The association between gene expression in CTCs and aggressiveness has been well reported in prostate cancer [122]. CTCs expressing high levels of *SPINK1* and *BIRC5* are associated with aggressive molecular subtypes [123] and castration-resistantance [124] of prostate cancer (Figure 2). The presence of *RRM2* and its regulated genes were also detected in CTCs of prostate cancer patients with enzalutamide resistance through scRNA-seq analysis (Figure 2) [125]. Among all detected genes from 77 prostate CTCs, an increasingly expressed 11-gene panel of the *RRM2* signature was significantly found in enzalutamide-resistant patients and associated with poor clinical outcomes in the validation cohort. It was consistent with previous studies that showed overexpression of *RRM2* in tumor tissue was a key driver of aggressive features and poor patient outcomes [126]. Metastasis gene signatures involved in the EphB2 and Src pathways, a downstream signaling pathway regulating epidermal growth factor receptor (EGFR) dynamics, were also identified by scRNA-seq analysis, in which the upregulation of several genes in the EphB2 and Src pathways was predominantly detected in CTCs with advanced malignancy (Figure 2) [127].

#### 3.1.3. CTC Subtyping Analysis in Other Biological Fluids

The power of scRNA-seq to reveal complexity and rare cell populations is not limited to blood samples. ScRNA-seq analysis in non-blood samples further explains the spatial metastatic site, the temporal difference during disease progression, and the patient-specific state, and is applied for discovering new markers of CTCs that might be hardly detectable in CTCs from blood samples [128]. A comprehensive atlas of CTCs in human body fluids can enhance our understanding of their role in therapeutically related chemotherapy resistance and tumor relapse.

The study of transriptomic profiles of CTCs derived from cerebrospinal fluid (CSF) samples of lung adenocarcinoma (LUAD) patients with leptomeningeal metastases revealed a different gene expression profile between CSF-CTCs and their matched blood-CTCs [129]. CSF-CTCs had a high expression of lung-specific genes, epithelial gene markers, and proliferation gene markers, but slightly found partial sets of mesenchymal- and CSC-related genes. Conversely, extracellular matrix (ECM)-related genes were typically upregulated in all patients. The upregulation of ECM might involve the generation of CTCs from the primary site or the driving survival mechanism of disseminating cancer cells in CSF.

Another study reported the scRNA-seq analysis of disseminated tumor cells (DTCs) in malignant pleural effusion samples from LUAD patients [130]. DTCs were subclassified into four groups according to gene expression profiles involving immune response, cell proliferation, apoptosis, and cell adhesion, respectively [130].

### 3.2. Revealing the Altered Molecular Pathways Underlying Cancer Progression for Druggable CTCs

Due to the fact that CTCs are the major player in cancer metastasis and represent cancer cells resistant to drug treatment, understanding the biology underlying these malignant characteristics will lead to the discovery of new targets for novel cancer treatments. The investigation of CTCS using scRNA-seq provides insightful information on molecular regulation during cancer metastasis and drug resistance. This section provides a comprehensive review of the CTC mechanisms investigated through scRNA-seq analysis and the promising treatments.

#### 3.2.1. Metastatic Mechanisms

The link between ECM genes in CTCs and metastasis was found in pancreatic cancer [131]. By using scRNA-seq, a wide range of proteinaceous stromal-derived ECM was highly enriched in CTCs from pancreatic cancer, which were rarely found in matching primary tumors. Among all ECM gene transcripts, the SPARC gene was remarkably overexpressed in all CTC samples. The knockdown of SPARC significantly suppressed cancer metastasis, which included the invasive and migratory properties observed in a mouse model.

ScRNA-seq analysis can potentially determine oncotargets, not only in CTCs but also in their blood tumor microenvironment. Brechbuhl et al. (2020) disentangle the scRNA-seq profile of intact CTCs isolated from cancer patients with active metastatic disease and local breast samples [92]. CTCs were subgrouped into CTC-1, established estrogen-responsive CTCs with highly proliferative features, and CTC-2, more quiescent cells with increased numbers of EMT-associated genes. Interactome and pathway analysis revealed that CTCs likely interacted with platelets through megakaryocyte interactions, through which several markers involved in activated platelets were increased. In addition, the PD-1/PD-L1 immune checkpoint pathway was predicted to have increased activation in peripheral lymphocytes from metastatic patients. These data suggested that CTCs participate in blood immune cells through receptor/ligand pairs that decrease immune surveillance.

Furthermore, scRNA-seq analysis reveals significantly heterogeneous populations of CTCs and crosstalk between CTCs and blood immune cells at single-cell resolution. Recent findings apply the scRNA-seq technology to investigate spatial transcriptomic changes of CTCs during hematogenous transportation in HCC [51]. The transcriptomic profiles of CTCs were different in that CTCs isolated from liver efferent vessels reflect their primary tumor heterogeneity, whereas CTCs in efferent vessels represented an adaptation of CTCs in the blood circulatory system. The gene sets identified in CTCs were predominantly associated with stress responses, cell cycle, and immune evasion signaling pathways. All of these biological processes aided the CTC’s survival against circulatory cytotoxicity. The chemokine CCL5 was a significant and important mediator of CTCs in immune evasion and consequently promoted CTC-mediated metastasis through the recruitment of regulatory T cells (Treg), as shown in Figure 2. Mechanistically, in vitro studies found that CTCs acquired the ability to attract immunosuppressive Treg cells via an exaggerated activation of p38-MAX-CCL5 signaling pathways. This finding provides the possibility of designing a novel anti-metastatic therapeutic option for HCC through immunotherapy targeting Treg and/or CCL5-positive CTCs.

#### 3.2.2. Drug Resistance Mechanisms

ScRNA-seq data of CTCs showed that non-canonical (nc)-Wnt signaling is activated in CTCs derived from prostate cancer patients given AR-targeted therapy [114]. Ectopic expression of the non-canonical ligand, WNT5A, diminished the anti-proliferative effect of AR inhibition, in which the expression of endogenous WNT5A was also increased upon treatment, and WNTA5 knockdown partially restored the sensitivity in drug-resistant cells. In this context, it has been found that canonical (c)-Wnt and nc-Wnt signaling activation played vital roles in the evolution of cancer stemness and regulated the expression of functional CSC markers (Figure 2) [132]. The c-WNT cascades directly promote the self-renewal of cancer cells and control their cell fate and function through transcriptional regulation. By contrast, nc-WNT was more associated with the maintenance of stemness, directional cell movement, and the survival and therapeutic resistance of CSC. Combinatorial treatment targeting both WNT signaling pathways could be a potential oncotherapy [133].

## 4. Future Perspectives and Conclusions

The evolving field of CTC research in cancer patients has recently been recognized as “the golden age of liquid biopsy” [134]. scRNA-seq holds great potential to revolutionize the fundamental understanding of CTC biology. Only a small number of all disseminated CTCs play a key role in metastasis [135,136,137,138]. The characterization of individual CTCs at high resolution provides valuable information for therapeutically targeting tumor metastasis and their interplay in transit and beyond.

Cancer cells require mobility and migratory ability to reach metastatic distant sites through the invasion of the EMT process [139]. This reversible state of the cancer cell phenotype leads to CTC formation and dissemination [140]. Recent studies of single-cell analysis have demonstrated the spatial heterogeneity of CTCs across different vascular sites [141]. CTCs exhibited phenotypic changes during dissemination in blood circulation. A majority of CTCs released from primary tumor sites had an epithelial phenotype, whereas CTCs collected after crossing microvessels predominantly expressed a mesenchymal phenotype. Finally, this EMT process was reversed toward the MET process for metastatic colonization.

Therapeutic targets of mesenchymal markers such as anti-N cadherin monoclonal antibodies have considerable clinical benefit for blocking local invasion and metastasis in androgen-dependent to castration-resistant prostate cancer [142]. Consistently, withaferin A has been shown to inhibit the vimentin cytoskeleton by inducing the degradation of vimentin in a breast cancer model [143]. Due to several complex networks regulating the EMT program, interference of Wnt [144], Notch [145,146], and Hedgehog [147] signaling cascades has been explored as potential therapeutic options in pre-clinical models. Furthermore, targeting the upstream modulator of EMT activation by EGFR inhibitors (AG1478) [148] and TGF-β inhibitors (Fresolumimab/GC1008) [149] effectively blocked the paracrine EMT-inducing signal.

CTCs travel in the circulation as a single or a cluster of different CTC phenotypes, in which the CTC clusters appear to have stemness and plasticity. These homo-clusters extraordinarily strengthen CTCs ability to form metastases through increased resistance to anoikis and shear stress [150]. Hence, the survival adaptation of CTCs to anoikis resistance can be targeted by blocking survival signals, rendering CTCs susceptible to apoptosis. The use of neurotrophic tyrosine kinase receptor (Trkb), a specific suppressor of caspase-associated anoikis, has been conducted by CEP-701 and CEP-2563 in clinical trials as a treatment for patients with advanced carcinomas (Figure 2) [151,152,153]. The inhibition of the pro-survival signaling pathway by PI3 kinase inhibitors (LY294002, PI103, and ZSTK474) had remarkable effects in xenograft models [154,155,156] and clinical studies [157] against a range of solid tumors and sarcomas.

Other therapeutic options involve the inhibition of host cellular environmental factors. CTCs can aggregate into a multicellular population, thereby enhancing immune escape [158]. During the transit in the vasculature, disruption of CTCs forming heterotypic clusters with host platelets is proposed by using anti-platelet agents. The prevention of tumor-cell-platelet interaction via dipyridamole and its analog, RA-233, was reported to provide an effective intervention in both in vitro and in vivo systems for the treatment of metastases [159]. Direct targeting of integrin αVβ3 expressed on tumor cells and inducing tumor-platelet aggregation is a promising anti-cancer and anti-angiogenic approach [160]. In a similar manner, scRNA-seq identifies a possible mechanism by which CTCs promote T-cell exhaustion through regulatory T cells and the PD-L1/PD-1 pathway. Such approaches as the blockade of checkpoints by anti-PD-1, anti-PD-L1, and anti-CTLA-4 have been appealing as the greatest option for restoration and recovery of T cell exhaustion [161]. From a practical standpoint, T-cell exhaustion remains a practical challenge for developing drugs targeted at their transcriptional machinery. With the transcriptional regulation involved in the regulation of exhausted T cells, the full picture of their gene regulatory networks needs to be explored. In particular, the interaction between CTCs and blood microenvironments differs greatly in different cancers and/or in different people. We expect that the scRNA-seq method for CTCs that accurately determines the precise molecular mechanism might confer superior advances in combinatorial cancer immunotherapies.

CTC culture, xenograft, and organoid are being developed as models for drug screening. Although it has some challenges to culture rare CTCs, up to 358 cell lines derived from CTCs have been successfully established for many cancer types [162]. In this regard, drug susceptibility has been tested in CTC models with a number of approved drugs. The efficacy of drugs to target CTCs was further investigated in several studies in vitro as well as in vivo. More recently, patient-derived organoid models have been introduced and have become mainstream drug screening platforms with high efficiency and sensitivity in preclinical research [163]. In contrast to the xenograft model, the organoid system can be expanded for continued culture and recapitulate its parental tumor characteristics, such as cellular heterogeneity, structure, and function. CTC-derived organoids have been successfully established in various cancer types, including colorectal [164], prostate [165], pancreatic [166], head and neck cancer [167], and soft tissue sarcoma [168]. This experimental model may lead to new strategies for investigating metastatic mechanisms for the identification of more effective and precise therapies.

ScRNA-seq of CTCs provides a wealth of information about their tumors of origin as well as their cellular fate. It is a potent technology with unbiased identification of CTCs that is useful for investigating transcriptional changes, monitoring the progression of disease, indicating appropriate treatments, and ultimately identifying precision therapeutic targets, which will improve cancer patient compliance and clinical outcomes.

Despite all these great promises, there are several challenges to scRNA-seq analysis of CTCs that limit its clinical application from benchtop to bedside. Dissecting CTCs at the single-cell level strongly depends on technological development, which includes the need to improve the low capture efficiency of extraordinarily rare CTCs, the possibility of non-single-cell resolution, and the high fidelity of the sequencing method. Furthermore, extensive bioinformatic tools developed for the interpretation of scRNA-seq data are needed. Bioinformatics analyses of scRNA-seq data are more complicated than bulk transcriptome analyses due to high background noise, batch effects, and technical errors. It Is anticipated that more capable bioinformatics analysis tools will be established and become accessible in the future. Furthermore, performing scRNA-seq for multiple samples is labor-intensive, and the cost remains substantially high. Fully automatic CTC isolation enabling fast sample processing and non-laborious sample preparation can result in high CTC recovery and high purity with intact cells [18,169]. Likewise, the availability of automated and customizable pipelines for large-scale scRNA-seq data analysis needs to be standardized.

Together, the continuing advancement of CTC liquid biopsy and its combination with scRNA-seq and/or muti-domain technologies will bring a new breakthrough in “translating cancer omics”, eventually benefiting cancer patients in ways never before possible.

## Figures and Tables

**Figure 1 ijms-24-12337-f001:**
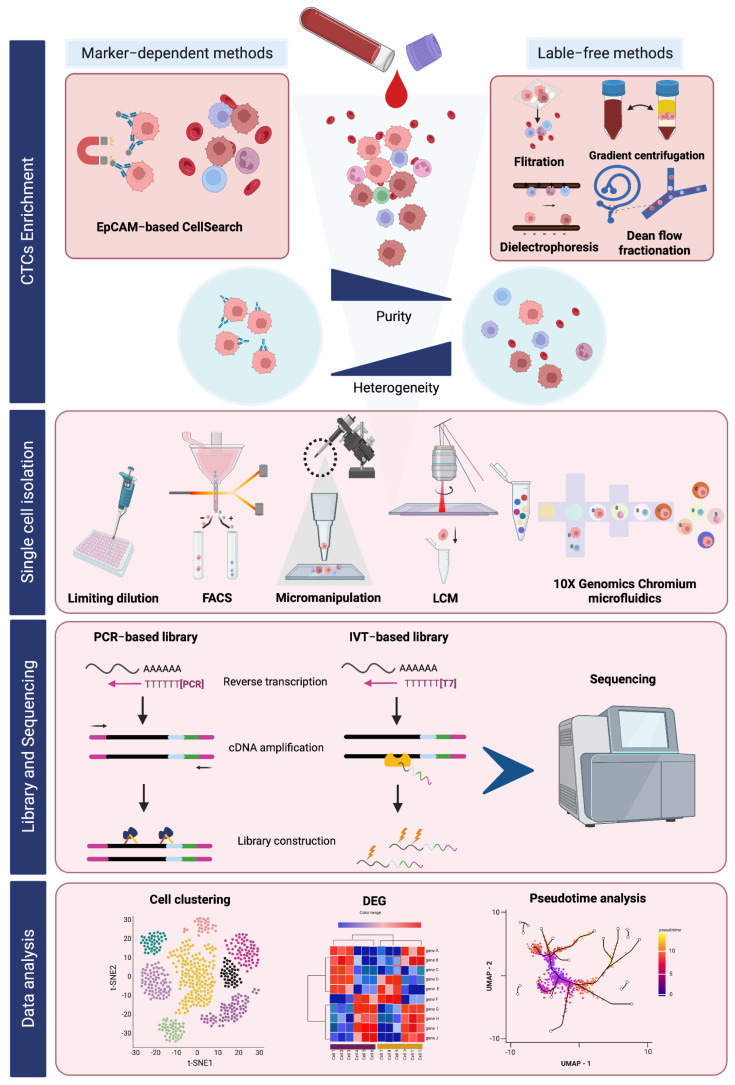
An overview of CTC enrichment and single-cell RNA sequencing workflow.

**Figure 2 ijms-24-12337-f002:**
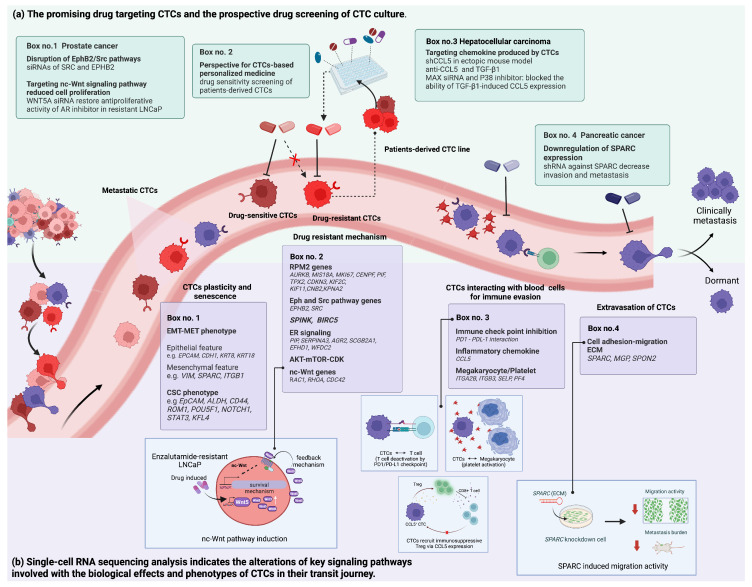
A summary finding of the scRNA-seq analysis of CTCs and their potential targets for clinical relevance. (**a**) The representative promising drug targets for CTCs that were disclosed by scRNA-seq technology and the prospective for personalized medicine of individual CTCs by drug susceptibility tests. (**b**) The transcriptional and phenotypic heterogeneity of tumor cells in the transit journey of CTCs. As the intermediator conduit to metastasis, CTCs possess different phenotypic characteristics, including EMT and CSC phenotypes, drug resistance, CTCs coupled with immune cells, and extravasating CTCs. Thus, gene expression profiling of CTCs could accomplish a complete understanding of the significance of CTC dissemination and hold remarkable promise for advancing precision cancer therapy.

## Data Availability

Not applicable.

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
