# Peer review of "Deciphering the Biology of Circulating Tumor Cells through Single-Cell RNA Sequencing: Implications for Precision Medicine in Cancer"

_ijms, 2023, doi:10.3390/ijms241512337_

Round 1
Reviewer 1 Report
In this manuscript Orrapin et al. describe the recent advances in circulating tumor cell single cell RNA sequencing and display their implications for precision medicine in cancer.
The authors show the technological difficulties to isolate and characterize CTCs as well as they point out the translational relevance of scRNA sequencing of CTCs.
Overall evaluation:
The manuscript is well constructed and raises an important and highly developing subject. However some of the sections require important improvement.
1. Introduction line 39: Authors should describe more in details EMT process and its role in tumor dissemination, significant papers should be cited here.
2. Page 3 line 111 – 140 – this fragment is repeated above
3. Page 4 section 2.1.3 – this section is out of topic, otherwise should be revised
4. Page 5 line 215-216. More details concerning results obtained from patient samples should be mentioned.
5. Page 5 line 227 “experiments showed that an affordable quality of individual cells”. Please specify : the recovery of CTCs, either the experiments considered patient CTCs or spiked cell line etc.
6. Page 6 line 250 is repeated twice
7. Page 8 lines 335-352. This section contains too many bioinformatics analysis, hardly to understand and not enough explication either the presented pipelines have been developed for single cell analysis.
8. Page 13: Figure 2 is incomprehensible. Is it a resume of literature ? If yes the references are not present. Does the mentioned cancer therapies concern all types of tumors, specific types ? What was the goal of authors to show this figure? Please clarify, add references if necessary, increase the police or present the same in form of table.
Minor editing of English language required
Author Response
We are grateful for the review and hope to have correctly addressed Reviewer’s comments. The reviewer has dedicated to providing valuable feedback on our manuscript. We have been able to incorporate the changes to reflect most of the reviewer’s suggestion. The substantial changes were made in revised manuscript and are highlighted in yellow.
Reviewer 1
Comments and Suggestions for Authors
In this manuscript Orrapin et al. describe the recent advances in circulating tumor cell single cell
RNA sequencing and display their implications for precision medicine in cancer. The authors show the technological difficulties to isolate and characterize CTCs as well as they point out the translational relevance of scRNA sequencing of CTCs.
Overall evaluation:
The manuscript is well constructed and raises an important and highly developing subject.
However, some of the sections require important improvement.
- Introduction line 39: Authors should describe more in details EMT process and its role in tumor dissemination, significant papers should be cited here.
Response: Thank you for this suggestion. We have, accordingly, done to emphasize this point in revised manuscript.
Revised in Page 1, line No. 39 – 47
The EMT process is the transdifferentiation of epithelial cells by which the cells lose their epithelial characteristics and acquire mesenchymal features [4]. Cancer cells hijack these dynamic changes in morphology and motility to conduct cancer migration, invasion, and intra- and extravasation into the circulatory system. The EMT process also activates several survival pathways to avoid anoikis and induce resistance to chemotherapy and physical stress [5, 6]. The EMT phenotype of CTCs is to survive in the bloodstream until undergoing mesenchymal-to-epithelial transitions (MET), and restoring their epithelial character to extravasate and colonize distant sites [7].
- Page 3 line 111 – 140 – this fragment is repeated above.
Response: The mistake has been corrected in revised manuscript
- Page 4 section 1.3 – this section is out of topic, otherwise should be revised.
Response: Section 2.1.3 of Laser capture microdissection (LCM) is included as a sub-topic of the “Single-cell isolation techniques” topic. In this section, we aim to highlight the techniques applied for isolating single cells of enriched CTCs. LCM is one of the methods frequently used to capture single CTCs based mainly on their morphology under microscopic observation. This method provides ultra-pure CTCs that are applicable for downstream molecular analysis, including sc-RNA seq.
- Page 5 line 215-216. More details concerning results obtained from patient samples should be mentioned.
Response: More results have been added, accordingly.
Revised in Page 2, line No. 255-261
Hydro-seq represents high CTC capture efficiency and reproducibility. The number of CTCs was recovered at a high cell capture efficiency of 72.85 ± 2.64% (mean ± SD) and gene expression profiles are comparable with the results from biological replicates of blood samples from the same patients. Additionally, a low percentage of cells expressing CD45 leukocyte markers and hemoglobin was found in the scRNA-seq data. The results from gene expression analysis confirm the efficacy of removing massive contamination of blood cells in a single-cell capture chamber.
- Page 5 line 227 “experiments showed that an affordable quality of individual cells”. Please specify: the recovery of CTCs, either the experiments considered patient CTCs or spiked cell line etc.
Response: Thank you for your suggestion. It would have been interesting to point out this aspect. We have added more details about the recovery and quality of CTCs after enrichment by their SCR-chip.
Revised in Page 2, line No. 267-272
The spiking experiments showed that an affordable quality of individual cells isolated from peripheral blood without leukocyte contamination were obtained by SCR-chip [41]. The CTC enrichment efficiency of an integrated microfluidic chip with low to high cell numbers (10 to 2000) of the MCF-7 cell line varies between 55%-93%, which is relatively dependent on the adjusted flow rate. The RNA yields and quality obtained from trapped single-cell CTCs are adequate for RNA sequencing and transcriptomic analysis.
- Page 6 line 250 is repeated twice.
Response: The mistake has been corrected in revised manuscript.
- Page 8 lines 335-352. This section contains too many bioinformatics analysis, hardly to understand and not enough explication either the presented pipelines have been developed for single cell analysis.
Response: In the revised manuscript, the paragraph has been rewritten to provide a clearer explanation.
Revised in Page 8, line No. 395 - 406.
Although many benchmarks for trajectory analysis have been developed, a typical workflow of these methods is designed into three main steps: (i) preprocessing data by filtering out the poor-quality cells and uninformative expressed genes, normalization to remove undesired technical or biological errors, and dimensionality reduction, (ii) clustering the cells by determination of global lineage structure, and (iii) pseudotemperal reconstruction by projection of respective cells onto their lineage individuals along time within one biological sample [95]. In a comprehensive statistical framework, the analysis uncovers multiple aspects, including (i) the genes in which expression is associated with programmed lineages along differentiation, (ii) differential gene expression between the transcriptional lineages over pseudotime, and (iii) a density or proportion of cells along lineage across conditions [96].
- Page 13: Figure 2 is incomprehensible. Is it a resume of literature ? If yes the references are not present. Does the mentioned cancer therapies concern all types of tumors, specific types ? What was the goal of authors to show this figure? Please clarify, add references if necessary, increase the police or present the same in form of table.
Response: Thank you for your suggestions. This figure illustrates the summary of key pathways collected from CTC scRNA-seq data. These pathways significantly involve the journey of CTCs from the initial tumor site to a distant organ. Furthermore, several studies have tested the function of important mediators in CTC survival. Therefore, we highlight the promising therapeutic targets for eradicating CTCs in this figure. The use of CTCs as a model for drug screening is another outstanding approach for drug repurposing and discovery, so we added the information in the figure and in the main text.
We made some changes to Figure 2 to incorporate your comments, and the figure caption has been reconstructed in the revised manuscript (Page 11, line No. 616-625). The bottom half of figure (b) represents the overall mechanistic hallmarks disclosed by scRNA-seq analysis of CTCs. We shortened the molecular mechanisms involved in the metastatic process of CTCs as well as the mediated pathway interacting with blood immune cells. Furthermore, some existing cancer drugs/therapies might inhibit the metastatic ability of CTCs specifically on the respective pathways, which are shown in the top half of figure (a).
Figure 2. A summary finding of the scRNA-seq analysis of CTCs and their potential targets for clinical relevance. a) The representative promising drug targets for CTCs, that were disclosed by scRNA-seq technology and the prospective for personalized medicine of individual CTCs by drug susceptibility tests. b) The transcriptional phenotypic heterogeneity of tumor cells in the transit journey of CTCs. As the intermediator conduit to metastasis, CTCs possess different phenotypic characteristics, including, EMT and CSC phenotypes, drug resistance, CTCs coupled with immune cells, and extravasating CTCs. Thus, gene expression profiling of CTCs could accomplish a complete understanding of the significance of CTC dissemination and hold remarkable promise for advancing precision cancer therapy.
Revised in Page 12-13, line no. 683-696
CTC culture, xenograft, and organoid are being developed as models for drug screening. Although it has some challenges to culture rare CTCs, up to 358 cell lines derived from CTCs have been successfully established from many cancer types [163]. In this regard, drug susceptibility has been tested in CTC models with a number of approved drugs. The efficacy of drugs to target CTCs was further investigated in several studies in vitro as well as in vivo. More recently, patient-derived organoid models have been introduced and have become mainstream drug screening platforms with high efficiency and sensitivity in preclinical research [164]. In contrast to the xenograft model, the organoid system can be expanded for continued culture and recapitulate its parental tumor characteristics, such as cellular heterogeneity, structure, and function. CTC-derived organoids have been successfully established in various cancer types, including colorectal [165], prostate [166], pancreatic [167], head and neck cancer [168], and soft tissue sarcoma [169]. This experimental model may lead to new strategies for investigating metastatic mechanisms for the identification of more effective and precise therapies.

Reviewer 2 Report
In the paper titled "Deciphering the biology of circulating tumor cells through single-cell RNA sequencing: Implications for precision medicine in cancer," the authors review the enrichment, single cell isolation, scRNA-seq, and bioinformatic analysis of circulating tumor cells. The topic is important in the field of biomedical and translational medicine. However, there are several limitations in the current manuscript.
1. On page 2, the authors mention that "CTCs are extremely rare, found with only one cell per billion blood cells in circulation, either as a single unit or multicellular grouping." This poses a significant challenge in capturing CTC cells through blood samples. However, in section 2.1, there are no further details provided regarding the enrichment process and the results obtained after enrichment. For instance, can the authors quantitatively describe the CTC proportions after enrichment for different cancer types? This critical step serves as the foundation for all subsequent procedures. Please provide more details on this aspect.
2. How can CTCs be obtained from such a small number of cells, even after enrichment? What features are used for FACS sorting in different cancer types? Please consider creating a summary table for the different methods of enrichment and isolation, highlighting the features utilized for different cancer types.
3. Given the rarity of CTCs, how can clustering be performed effectively? Most clustering algorithms exhibit poor performance in detecting/clustering rare cell subpopulations. Please provide additional details on this issue.
4. The differential expression analysis is limited to small cell numbers (<100). Considering the rarity of CTCs, how can DEG analysis be performed effectively when they are so rarely captured?
The English is okey but minor edits needed.
